# Fundamental Properties and Clinical Application of 3D-Printed Bioglass Porcelain Fused to Metal Dental Restoration

**DOI:** 10.3390/ijms24087203

**Published:** 2023-04-13

**Authors:** Yangan Yun, Hyeon Kang, Eun-Chae Kim, Sangwon Park, Yong-Seok Lee, Kwidug Yun

**Affiliations:** 1Department of Prosthodontics, School of Dentistry, Chonnam National University, 33 Yongbongro, Bukgu, Gwangju 61186, Republic of Korea; 2Ace Dental Clinic, 56, Namak 3-ro, Samhyang-eup, Muan-gun 58567, Republic of Korea; 3Department of Dental Laboratory Technology, Gwangju Health University, 73, Bungmun-daero 419beon-gil, Gwangsan-gu, Gwangju 62287, Republic of Korea; 4Department of Mechanical Engineering, Myeongji University, Yongin 17058, Republic of Korea

**Keywords:** 3D printing, SLM, margin fitness, dental restoration

## Abstract

The purpose of this study is to evaluate the mechanical properties and clinical fitness of 3D-printed bioglass porcelain fused to metal (PFM) dental crowns. To evaluate the mechanical properties, tensile strength, Vickers microhardness, shear bond strength, and surface roughness tests of the SLM printed Co-Cr alloy was conducted. A right mandibular 1st molar tooth was prepared for a single dental crown (*n* = 10). For a three-unit metal crown and bridge, the right mandibular first premolar and first molar were prepared. Bioglass porcelain was fired to fabricate PFM dental restorations. A clinical gap was observed and measured during each of the four times porcelain was fired. A statistical analysis was conducted. The SLM technique showed the largest statistically significant tensile strength and a 0.2% yield strength value. The milling technique had the lowest statistically significant compressive strength value. The shear bond strength and surface roughness showed no statistically significant difference between the fabricated method. There was a statistically significant change in marginal discrepancy according to the porcelain firing step. The casting technique showed the greatest statistically significant margin discrepancy value. The SLM method showed better fitness than the traditional casting method and showed better mechanical properties as a dental material.

## 1. Introduction

Porcelain fused to metal (PFM) restoration refers to porcelain restoration supported by a metal substructure, and it is one of the most commonly used restorations in dentistry [1]. PFM restoration is made of non-precious metals, such as Ni-Cr or Co-Cr alloys, instead of noble metal alloys due to their high cost. These non-noble metal alloys are affordable, exhibit high strength, modulus of elasticity and corrosion resistance, and therefore represent perfect candidates for metal substructures [1]. Especially, Co-Cr alloy has been used primarily as a metal substructure in removable partial dentures in the past. Currently, its implementation as a metal substructure in PFM restoration is increasing due to fewer side effects and higher internal corrosion resistance than Ni-Cr alloy [2,3]. Although metal coping structures using Co-Cr alloy have been mostly manufactured with the conventional lost-wax casting technique, the high fusion temperature of Co-Cr alloy is a hindrance to successful fabrication, in addition to inevitable errors associated with serial fabrication procedures. 

Computer-Aided Design/Computer-Aided Manufacturing (CAD/CAM) techniques play a major role in the production of biomechanical appliances and dental prosthodontic restorations [4,5]. Currently, dentists face challenges in providing individual restorations and prosthodontic reconstructions customized to each patient, which entails recreating complicated anatomic forms [6]. Although it may be possible to manufacture complex restorative structures using subtractive manufacturing techniques of multi-axial CAD/CAM milling equipment, it is very time-consuming and generates unnecessary material waste; often, the final product lacks precision depending on the size and shape of the milling equipment [7,8,9]. However, additive manufacturing, first suggested in the 1980s, represents the opposite concept of subtractive manufacturing. Recently, it has been studied in various fields, including aerospace technology and precision part manufacturing. It is expected to lead to phenomenal changes in manufacturing industries. The American Society for Testing and Materials (ASTM) defines additive manufacturing as a set of procedures combining materials to design the desired object, which generally results in the accumulation of specific layers of materials. Additive manufacturing procedures enable the precise production of various complicated restorations and structures in the field of dentistry and surgery [10,11].

Additive manufacturing can be classified according to methods such as material procurement, sources of energy, and accumulation volume. It is also largely categorized by powder bed systems, powder feed systems, and wire feed systems. Selective laser melting (SLM) is one of the metal accumulation techniques that belong to the powder bed system, which generates structures by directly irradiating metal powder with a high-powered laser beam, resulting in fusion between metallic particles. SLM has been used to make metal copings as well as metal substructures of partial dentures in previous studies. This technique yielded satisfactory results, with the final product having superb mechanical and chemical characteristics, such as high density, strength, and corrosion resistance [12,13,14].

Co-Cr alloys have been manufactured using both subtractive and additive methods via CAD/CAM technology. Co-Cr alloys can be developed easily and rapidly using additive manufacturing without the need for complex procedures, unlike conventional lost-wax casting techniques [4,10,15]. However, the study of Co-Cr alloy using the SLM technique is still in the initial stages. In addition, the disadvantages of current PFM restorations include the high risk of chipping and the fracture of the porcelain veneer, which is an important factor in determining the clinical success of the restoration [16,17]. Since Co-Cr has been mainly used in removable prostheses in the past, there are relatively few studies on fixed prostheses made in combination with ceramic materials [18,19,20,21,22,23]. Hence, the comparison of mechanical properties and the bonding strength of metal and porcelain of the restorative structure made with conventional manufacturing techniques should be evaluated before the clinical application of Co-Cr alloy using the SLM technique. However, the bonding strength of Co-Cr produced by the SLM method so far is a laboratory experiment that is different from the clinical situation. In addition, porcelain firing is rarely performed in accordance with the clinical situation. Instead of porcelain firing, thermocycling or a firing procedure without porcelain is performed [24]. However, it is very different from the actual clinical situation. Therefore, in this study, single crowns and three-unit bridges were used in a very similar way to clinical situations.

Marginal fit is one of the necessary requirements for the clinical success of dental restorations. Good marginal fit is essential for the long-term success of dental restorations, and the presence of any errors can render harmful effects on the abutment tooth and the supporting periodontal tissue. An inadequate internal fit could cause lead to a lack of resistance of the metal–ceramic restoration, the loss of axial retention, and decreased fracture resistance. According to the definition by Holmes, the internal gap refers to the vertical measurement from the axial wall of the abutment tooth to the internal surface of the restorative structure, and if the same measurement occurs at the marginal area, it is defined as a marginal gap [25]. Mclean confined a marginal gap of 120 μm as a clinically acceptable range [26]. Many studies have been conducted to investigate the marginal fit of conventional cast manufacturing methods of Co-Cr alloy metal–ceramic restorations [27,28]. However, the study determining the marginal fit of Co-Cr alloy restoration obtained by the SLM technique is still limited. Additionally, metal–ceramic restorations go through a series of porcelain firing procedures, which result in distortion of metal coping due to repeated heat treatments, resulting in a poor marginal fit. However, the study of this topic is inconclusive [29,30,31]. Specifically, there are no papers that have conducted experiments in the form of bridges that are actually used in clinical practice rather than specimens.

Therefore, the purpose of this study was to compare the mechanical properties of Co-Cr alloy generated using SLM, milling, and casting techniques for comparative evaluation of the fitness of the PFM restoration of each of the techniques mentioned above.

## 2. Results

### 2.1. Mechanical Properties

#### 2.1.1. Tensile Strength of Metal Alloy

The results of tensile strength according to casting, milling, and SLM manufacturing techniques are presented in Figure 1. The SLM technique showed a higher mean maximum tensile strength compared with the milling and casting techniques, which was statistically meaningful (*p* < 0.05). When comparing the elongation amongst the groups, the data did not show statistically meaningful differences (casting technique: 5.41 ± 1.81%, milling technique: 9.46 ± 3.04%, SLM technique: 8.69 ± 1.69%) (*p* > 0.05) (Figure 2). The 0.2% yield strengths (casting technique: 716.71 ± 50.03 MPa, milling technique: 512.65 ± 86.42 Mpa, SLM technique: 879.30 ± 34.32 Mpa) turned out to be significantly higher in the order of SLM, casting, and milling techniques, respectively (*p* < 0.05) (Figure 2).

#### 2.1.2. Vickers Microhardness

Figure 3 shows the results of the Vickers microhardness test according to the casting, milling, and SLM techniques (Figure 3). The values of the mean and standard deviation of Vickers microhardness in the SLM, milling, and casting groups were 441.97 ± 16.12 kgf/mm^2^, 388.57 ± 19.41 kgf/mm^2^, and 431.76 ± 12.85 kgf/mm^2^, respectively. The mean value of Vickers microhardness was lower with the milling technique than those of the casting and SLM techniques, which was statistically significant (*p* < 0.05).

#### 2.1.3. Shear Bond Strength of Porcelain vs. Metal Alloy

Figure 4 shows the result of the shear bond strength test between Noritake super porcelain Ex-3 porcelain and the Co-Cr alloy specimens made via casting (24.28 ± 2.32 MPa), milling (20.66 ± 1.45 MPa), and SLM (22.55 ± 4.63 MPa) techniques (Figure 4). The difference in the shear bond strength of the three groups was not statistically significant (*p* > 0.05).

Figure 5 illustrates the porcelain attachment fracture surface at ×200 magnification with SEM after the shear bond strength test (Figure 5A–C). All of the specimens showed mixed adhesive failure (between fracture and adhesion surfaces) and cohesive failure (fracture inside the porcelain). The light gray area of the SEM image represents the residual opaque porcelain after the porcelain fracture, and the dark gray area denotes the Co-Cr alloy structure area (Figure 5D). Figure 5E shows the area fraction of adherence porcelain (AFAP) (%) value obtained from Si (silicone) EDS atomic analysis. AFAP (%) values were as follows: casting (74.22 ± 14.90%), SLM (73.85 ± 22.17%), and milling (60.86 ± 22.84%) techniques in the order from high to low but did not appear to be statistically significant.

#### 2.1.4. Metal Surface Roughness

Figure 6 shows the result of the surface roughness test (Ra) according to casting, milling, and SLM techniques measured under four different conditions: (1) raw, (2) raw + sandblasting treatment, (3) polishing, and (4) polishing + sandblasting treatment (Figure 6). The surface roughness measured under raw conditions was as follows: milling (0.37 ± 0.02 μm), SLM-bur (1.73 ± 0.21 μm), casting (1.91 ± 0.11 μm), and SLM (3.13 ± 0.49 μm) in the order of roughness to smoothness. The SLM specimens showed the highest value that was statistically meaningful, and the milling specimens showed the lowest statistically significant value (*p* < 0.05). The surface roughness was still consistent after the sandblasting treatment of the surface under raw conditions with milling (1.03 ± 0.17 μm), SLM-bur (1.95 ± 0.29 μm), casting (2.29 ± 0.24 μm), and SLM (3.31 ± 0.42 μm), which increased in roughness after the specimens were sandblasted. The surface roughness measured under polished conditions was as follows: milling (0.17 ± 0.01 μm), SLM (0.22 ± 0.04 μm), and casting (0.27 ± 0.02 μm). Casting specimens showed a higher surface roughness compared with milling specimens, which was statistically meaningful (*p* < 0.05). The surface roughness measured via sandblasting after polishing was as follows: milling (0.83 ± 0.14 μm), SLM (0.77 ± 0.10 μm), and casting (0.87 ± 0.24 μm), but was not statistically meaningful (*p* > 0.05).

#### 2.1.5. Metallurgic Structure

Figure 7 shows the metal surface structure of Co-Cr alloy specimens manufactured via casting, milling, and SLM techniques. (Figure 7). A significant difference was found on the surface metal structure in terms of the manufacturing techniques. The Co-Cr alloy specimens fabricated via the casting technique showed characteristic dendritic-like shapes. The Co-Cr alloy specimens fabricated with the milling technique showed an island shape. The Co-Cr alloy specimens designed with the SLM technique showed a layered cladding shape at ×100 and ×500 magnification and a fine surface along the inner border with apparent metal fusion at ×2000 magnification (Figure 7).

### 2.2. Dental Restoration Fitness Test

#### 2.2.1. Marginal Fit

The marginal fitness of all groups according to the porcelain firing stages of Co-Cr alloy coping was classified according to firing cycle steps, and the mean and the standard deviation were computed (Table 1).

The marginal fit showed increasing marginal discrepancy as the porcelain firing stages progressed. Instead, all groups showed a significant difference in the marginal discrepancy between the initial step and after the glazing step (*p* < 0.05) (Table 1).

The marginal fit according to casting, milling and SLM manufacturing techniques after the glazing step suggested that in the case of milling and SLM, the marginal discrepancy values were not statistically meaningful (*p* > 0.05). However, in the case of the casting method, the mean value of the marginal discrepancy was greater when compared with the milling and SLM methods, which was also statistically meaningful (*p* < 0.05) (Table 1).

The marginal discrepancy in the three groups significantly increased as the porcelain firing stages progressed (Table 1). (*p* < 0.05). The mean marginal gap values for the casting group were 81.05 μm, 88.04 μm, 91.65 μm, 96.09 μm, and 100.07 μm after the initial, oxidation, opaque, dentin, and glazing firing steps, respectively. The mean marginal gap values for the milling group were 67.55 μm, 75.68 μm, 79.80 μm, 82.05 μm, and 86.59 μm at the same firing stages. The mean marginal gap values for the SLM group were 71.62 μm, 77.63 μm, 83.76 μm, 86.45 μm, and 92.64 μm at the same firing stages. In the casting, milling, and SLM groups, the marginal discrepancy values increased in the order of the milling, SLM, and casting groups, which was statistically significant (*p* < 0.05). The mean marginal gap value was 91.38 μm for the casting group, 78.33 μm for the milling group, and 82.42 μm for the SLM group. The marginal gap after the completion of the glazing process is the most important value because the metal–ceramic restoration is used after the completion og porcelain firing. The marginal gap values after the completion of the glazing process were 100.07 μm for the casting group, 86.59 μm for the milling group, and 92.64 μm for the SLM group. When comparing the mean value of the marginal discrepancy between the single coping group and the bridge coping group according to the manufacturing techniques, all techniques showed higher marginal discrepancy values in the bridge coping groups than in the single coping groups, which was statistically meaningful (*p* < 0.05).

#### 2.2.2. Internal Fit

The mean and standard deviation of the internal fit were computed in all groups after the porcelain firing process was completed (Table 2). The values were arranged according to the manufacturing techniques, which were analyzed statistically. The axial internal gap was significantly lower in the milling than in the casting and SLM methods (*p* < 0.05), and the occlusal internal gap was significantly higher in the SLM than in the casting and milling methods (*p* < 0.05). When the axial and occlusal internal gaps were combined, the internal gap was higher in the order of the milling, casting, and SLM groups, and the milling and SLM groups showed statistically meaningful differences (*p* < 0.05).

For a three-unit bridge, the axial internal gap was lower in the milling group than in the casting and SLM groups, which was statistically meaningful (*p* < 0.05), and the occlusal internal gap was higher in the SLM group than in the casting and milling groups, which was statistically meaningful (*p* < 0.05) (Table 3). With respect to the combined axial and occlusal internal gap, the internal gap increased in the order of the milling, casting, and SLM groups (*p* < 0.05). When comparing the mean internal gap values of the single coping groups and bridge coping groups according to the manufacturing techniques, in all manufacturing techniques, the bridge coping groups showed higher internal discrepancy values than the single coping groups (*p* < 0.05).

## 3. Discussion

In this study, mechanical properties and restoration fitness tests were conducted with a Co-Cr alloy produced via additive, subtractive, and casting manufacturing techniques. Although the mechanical properties and fitness have been evaluated using additive manufacturing techniques in various studies recently, the study of Co-Cr alloy specimens is still in the beginning stage [18,20,21,22,23,24]. This study investigated the unique traits of metal alloys compared with the mechanical properties of Co-Cr alloy specimens fabricated using SLM, which is an additive manufacturing technique, with specimens made by conventional casting and subtractive manufacturing techniques. In addition, the restoration fit was evaluated by fabricating a PFM crown coping in a similar environment under actual clinical conditions created by executing porcelain build-up.

In the tensile strength test, ultimate tensile strength, elongation, and 2% yield strength were measured and classified according to the groups and specimens meeting the mechanical trait standards of ISO 22674:2006 in this study. The SLM technique showed higher UTS and 2% yield strength than casting and milling techniques, which was statistically meaningful (*p* < 0.05). These results are consistent with the current studies, showing that the Co-Cr alloy manufactured via SLM has improved mechanical traits compared with the conventional manufacturing techniques [8,32,33]. The grain size of the metal alloy is closely correlated with the nucleation rate, which is determined by the degree of supercooling [34]. As the specimens manufactured with the SLM technique exhibit substantially higher differences in temperature compared with those involving the casting technique (casting = 1500 °C, SLM = 1800 °C), it is expected that the grain size is significantly smaller in specimens exposed to the SLM technique. As shown in this study, the SEM images of the Co-Cr alloy specimen showed dendritic and island shapes with casting and milling techniques, respectively, whereas the SLM technique yielded fine grains. Casting technique specimens can be divided into dentritic and interdendritic structures, which separate metal solutes into two domains, resulting in the degradation of mechanical traits. The milling technique also showed a surface metal structure associated with the precipitated island-shaped phase instead of a monophasic structure. Similarly, the Co-Cr alloy specimens fabricated with casting and milling techniques also contained a precipitated phase. Conversely, the Co-Cr alloy specimens obtained via the SLM technique showed a surface structure with fine grains, which may result in more favorable mechanical traits. The distinction in the surface metal structures indicates a difference in solidification and thermo-mechanical procedures, which is consistent with previous studies demonstrating that specimens obtained via the casting and milling techniques carry a dendritic microstructure, whereas specimens prepared via the SLM technique showed a uniform and fine-grain microstructure [34]. The mechanical traits of the Co-Cr alloy are also affected by the composition of metallic phases, in which case, the monophasic composition improves the machinability of the metal alloy. The Co-Cr alloy undergoes a transformation from the FCC phase at high temperature to an HCP phase at low temperature, and in the case of the SLM technique, the specimens are processed via rapid solidification, which results in an alloy consisting mostly of the FCC phase [34]. Therefore, the SLM technique specimens show better machinability than the casting technique specimens.

The SLM technique specimens had the highest Vickers microhardness value at (441.97 kgf/mm^2^), which is consistent with recent studies [8]. The Vickers microhardness values of the casting and SLM techniques were higher than that of the milling technique, which was statistically meaningful. Recent studies indicate that the grain size at the time of manufacture is distributed more finely in the SLM technique, which enhances mechanical traits, and residual strength generated during fabrication appears to contribute to higher compressive hardness [8,34].

The results of the surface roughness test showed that the specimens under the raw conditions exhibit a rough surface in the order of milling > casting = SLM-bur > SLM. Metal additive manufacturing contains numerous metal pearls on the surface naturally, which requires finishing with a rotary cutting machine in order to build up an appropriate porcelain superstructure. After the finishing procedure, the surface roughness of the SLM specimens did not show a statistically meaningful difference compared with the casting technique specimens (*p* > 0.05), which coincided with the clinical appearance after the sandblasting processing before porcelain firing. Appropriate surface roughness increases the bonding surface with the porcelain superstructure, which possibly results in increased bond strength. The milling and casting techniques yielded statistically significant differences in surface roughness after polishing *(p* < 0.05). At the same time, the SLM technique specimens processed with surface polishing similarly did not show statistically meaningful differences compared with milling technique specimens *(p* > 0.05). Thus, it can be inferred that the polishing step transforms the macroscopic surface roughness of the SLM technique specimens to that of the milling technique. In addition, when the specimens were processed via sandblasting after polishing, the difference in surface roughness due to the manufacturing technique was not significant (*p* > 0.05). In brief, in the case of the SLM technique, the metal alloy surface under raw conditions contained multiple metal pearls generated during repeated melting and cooling procedures, but after appropriate surface finishing, the specimens showed similar surface roughness as those of the casting and milling techniques, which rendered appropriate surface conditions for the build-up of the porcelain superstructure.

Alloy manufacturing using other techniques increases the differences in the surface shape as well as changes in surface oxide, which affect the strength of the metal–porcelain bond [35]. In this study, the shear bond strength did not show a statistically meaningful difference when porcelain was fused to the Co-Cr alloy specimens via three different techniques (*p* > 0.05). The bond strength between the metal and porcelain is affected by the difference in mechanical and chemical bonding and thermal expansion coefficient [36,37]. According to the results of this study, the lack of difference in the shear bond strength due to the manufacturing techniques means that these complex factors do not contribute to the statistical differences. Likewise, although the result of AFAP showed a residual porcelain attachment rate based on the following order (high to low): casting (74.22 ± 14.90%), SLM (73.85 ± 22.17%), and milling (60.86 ± 22.84%), it was not statistically meaningful (*p* > 0.05). The AFAP of SLM and casting specimens was almost equal, which indicates that it is appropriate for clinical application since it suggests that the metal–porcelain bond strength also shows an equivalent performance.

In this study, the marginal fit was measured during a series of porcelain firing procedures of Co-Cr alloy coping manufactured using casting, milling, and SLM techniques. Accordingly, as the porcelain firing procedure progressed, the marginal gap increased little compared with the previous stage. Comparing the marginal discrepancy of the first stage, the ‘initial stage’, and the final ‘glazing stage’, all of the Co-Cr alloy coping specimens manufactured with casting, milling, and SLM techniques showed statistically meaningful differences (*p* < 0.05) due to the accumulation of the marginal distortion of the Co-Cr alloy coping generated during each porcelain firing stage. Although it is suggested that the clinical objective is to maintain the marginal gap between 25 and 40 μm, according to American Dental Association (ADA) specification No.8, this is difficult to achieve. McLean studied the marginal fit of about 1000 PFM crown restorations and suggested that a marginal discrepancy of about 120 μm is the clinical threshold [25]. This finding indicates that the marginal discrepancy generated after a series of porcelain firing procedures in this study is within the clinically acceptable scope. The marginal discrepancy values generated during the porcelain firing procedures of this study tend to be higher than those of previous studies, which executed porcelain firing repeatedly without the porcelain build-up [38]. It appears that as the temperature drops during the porcelain firing thermal treatment procedure, additional distortion in marginal fit occurs due to the differences in the coefficient of thermal expansion between the metal coping and the porcelain superstructure.

Previous studies have reported that the marginal discrepancy occurs due to the release of residual stress generated during the porcelain thermal treatment, which is the first stage of porcelain firing [30,39]. According to Papazoglou et al., statistically meaningful distortion occurs during the first step of the porcelain firing thermal treatment, whereas Li Zeng et al. reported that the increase in marginal discrepancy was not statistically significant [38,40]. In this study, although the mean marginal discrepancy during the first porcelain firing heat treatment did not show a statistically meaningful difference, a clear discrepancy was observed when the marginal discrepancy was compared before and after the firing. The Noritake Super Porcelain EX-3 used in this study is high-fusing dental porcelain that requires an increase in temperature as high as 1000 °C during the firing cycle. The decline in the alloy under this high temperature can induce distortion in the metal coping, which subsequently, under repeated thermal treatments at high temperatures, results in marginal distortion [41,42]. In conclusion, even though there was no significant marginal discrepancy generated during the first stage of porcelain firing, it can be assumed that the marginal distortion of the actual metal coping accumulates.

The casting method showed higher mean marginal discrepancy compared with the milling and SLM groups in all stages of the porcelain firing procedures, which showed statistically meaningful differences (*p* < 0.05). Even though milling showed a lower mean marginal discrepancy in all stages of porcelain firing, it was not statistically meaningful (*p* > 0.05). Örtorp et al. reported that when the marginal fit of Co-Cr alloy coping manufactured with casting, milling, and SLM techniques was compared, the marginal gap increased in the order of laser sintering, casting, and milling techniques [43]. Additionally, Quante et al. reported on the marginal fit when using a silicone replica, wherein the marginal discrepancy increased in the order of casting, milling, and laser sintering techniques [44]. The result of this study was inconsistent with those of previous studies, which can be attributed to the difference in the accuracy of the CAM unit during milling, a lack of heat generation in the manufacturing environment, differences in the technical expertise of the laboratory technician when using the complex procedures of casting using the lost-wax technique, and differences in the laser beam conditions and layering systems of SLM. However, it is difficult to create a detailed internal shape using the milling technique, and this entails complex restoration manufacturing procedures. In addition, the milling technique decreases precision due to the fast wear of the cutting instrument during milling and the significant waste of materials during the manufacturing procedures.

The SLM technique showed a higher internal gap compared with the other groups prepared using different techniques, which was consistent with previous studies [45,46]. It can be inferred that the difference in the accuracy of the equipment used and the distortion due to the heat-generating environment during manufacturing contributed to this result. Comparing the coping manufactured using the SLM technique and the casting technique, the internal fit of the SLM did not show statistically meaningful differences in terms of casting axially, but the occlusal portion showed a statistically meaningful difference (*p* < 0.05), which is contrary to the previous result that shows a better marginal fit of SLM. When evaluating the SLM technique specimens compared with the casting technique specimens, the internal gap decreases from the occlusal to the marginal part due to the accompanying cervical contraction of the metal alloy manufactured using the SLM technique in previous studies [38]. This result may be attributed to repeated high heat and rapid cooling treatments during the preparation of the specimen for the SLM technique, during which the metal coping with a supporter designed in the occlusal direction increases distortion due to the heat in the marginal portion, which is relatively far from the supporter. As a result, contraction in the marginal part is more prominent than in the occlusal part, and when measuring the internal fit, the marginal contraction restricts the applied silicone impression to the occlusal part, resulting in a high occlusal gap. For the milling group, higher occlusal gaps than axial ones were explained by the reduced scanner accuracy in occlusal areas, which have limited access for the milling burs [47]. Increased gaps are not desirable and result in increased cement thickness; these may also interfere with the accurate fit of a restoration. However, the internal gap value in this study exhibited a range of 167.86 μm to 189.26 μm. According to the previous study, the acceptable clinical occlusal gap range is 100 to 200 μm [46]. Thus, it is within a clinically acceptable internal gap range.

In this study, in all groups of casting, milling, and SLM technique specimens, the mechanical traits and marginal fit showed different results, and the null hypothesis was rejected. In the case of the SLM technique, it is possible to make restorations with complex forms, generate results without voids, and create Co-Cr alloy copings with better mechanical traits than those made using conventional techniques. Thus, a Co-Cr alloy coping manufactured with the SLM technique can be used to maintain long-term marginal soundness and is deemed appropriate for clinical application. Accordingly, the fitness of restoration and mechanical traits were excellent when fabricating PFM restorations via the SLM technique. Even though the fabrication of restorations using additive manufacturing techniques is limited by the exorbitant price of the equipment, the results of this study suggest the significant potential for dental applications in the future.

## 4. Materials and Methods

### 4.1. Mechanical Properties

All of the specimen compositions for SLM, milling, and casting techniques follow the ASTM F-75 standard so as to minimize the errors due to compositional differences in the materials (Table 4) [45]. The specimen preparation for the SLM technique was conducted using a 3D printer (ProX^®^ DMP 100, 3D systems Inc., Rock Hill, SC, USA), and SLM was used under 50 μm of Co-Cr metal power with the following parameters: P = 200 W, λ = 1070 nm, layer thickness = 20 μm, and speed = 40 mm/s. The specimen preparation in the milling technique was carried out using a milling machine (Arden 5X-WM, TPS Korea LTD, Gwang-Ju, Republic of Korea) and a Co-Cr Bar measuring 75 mm and weighing 4 kg apiece (R516045, Remelt Sources Inc., Cleveland, OH, USA), which was used for subtractive manufacturing. The specimen preparation for the casting technique was conducted by creating resin patterns with a 3D resin printer (Meg printer, Megagen, Dae-gu, Republic of Korea) using the Digital Light Processing (DLP) method, followed by burnout and casting procedures.

The Co-Cr alloy specimens used for the tensile strength test were formatted with a dumbbell shape in accordance with the ISO 22674;2016 standard and designed using a CAD program (AutoCAD^®^, Autodesk, CA, USA), which was then converted to an STL file format. Each of the 5 specimens was prepared using SLM, milling, and casting techniques [48]. The prepared dumbbell-shaped specimens were subjected to a tensile strength of 10 mm/min crosshead speed on the universal testing machine (AG-Xplus 50 kN, Shimadzu, Kyoto, Japan), and each specimen was recorded at the time of fracture.

The Co-Cr alloy specimens used for the Vickers microhardness test (disc-shaped specimen measuring 10 mm in diameter and 4 mm in height) were designed using CAD (AutoCAD^®^, Autodesk, CA, USA). The surface hardness of the specimens in all groups was measured with a micro-Vickers durometer (TUKON-1202, Wilson, IL, USA) and the hardness measurements were made below 1 kgf for a duration of 10 s. After removing the load, the diagonal indentation was observed with a microscope attached to a hardness tester, which was used to calculate the Vickers microhardness (*n* = 12).

All of the Co-Cr alloy specimens used in the shear bond strength experiment were polished on the surface to which the porcelain was attached using water spray. Next, the specimens were sandblasted with Al_2_O_3_ particles measuring 125 μm on the surface. Porcelain was stacked up to 8 mm in diameter and 4 mm in height. Noritake Super Porcelain EX-3 (Kuraray Noritake Dental Inc., Tokyo, Japan) was used for the porcelain build-up, and the porcelain furnace (Austromat 3001, DEKEMA, Freilassing, Germany) was used in accordance with the manufacturer’s instructions. The shear bond strength was measured using a universal testing machine (Model 1125, Instron, Canton, MA, USA) under the given load of testing speed at 1.0 mm/min until each specimen was fractured. To quantify the area fraction of adherence porcelain (AFAP), the specimen surface was studied in the center to determine the content of Si using SEM (SEM; scanning electron microscope, SNE-4500M plus, Dae-duk image, Dae-jeon, Republic of Korea) attached with EDS (EDS: energy-dispersive X-ray spectrometer, ESPRIT Compact, Brukernano GmbH, Berlin, Germany) at 200 x magnification. The atomic percentage of silicone (Si) was tested after sandblasting with 125 μm Al_2_O_3_, thermal oxidation (Sim), the application of opaque porcelain (Sio), and the elimination of porcelain from the specimens (Sif). The AFAP was estimated as AFAP = (Sif − Sim)/(Sio − Sim) (*n* = 5) [49].

The Co-Cr alloy specimens used for the surface roughness test measured 10 mm in diameter and 4 mm in height (*n* = 10). The surface roughness was measured under the following 4 conditions: (1) raw, (2) raw + sandblasting treatment, (3) after polishing, and (4) polishing + sandblasting treatment. In the surface roughness test of the specimens under the raw and raw + sandblasting treatment condition, the outer surfaces of the raw specimens made with the SLM technique were not treated entirely compared with the specimens obtained from the casting and milling techniques that were finished using a bur. Therefore, the surface roughness was measured in the following two groups: (1) specimens whose outer surfaces were depleted of metal pearl using stone bur (SLM-bur) and (2) specimens whose outer surfaces were not removed (SLM). All of the specimens were cleaned ultrasonically for 10 min using acetone, ethyl alcohol, and distilled water prior to the surface roughness test. The surface roughness of the specimens in all groups was measured using a 2D contact stylus profilometer (DIAVITE DH-8, DIAVITE AG, Feldstrasse, Switzerland).

To observe the metal structure, the specimen polished with No. 2000 SiC abrasive paper under water spray was photographed with SEM (SEM: scanning electron microscope, SNE-4500M plus, Dae-duk image, Dae-jeon, Republic of Korea). The specimens from all groups were observed at ×100 and ×500 magnification, and in the case of the Co-Cr alloy specimen made using the SLM technique, the surface observation was made at a ×2000 magnification for clearer observation of the surface structure.

### 4.2. Dental Restoration Fitness Test

Right mandibular first molars were selected from the standard resin tooth model (Dental model, NISSIN, Kyoto, Japan), which were then prepared by reducing by 1.5 mm on the labial, axial, and occlusal surfaces with a deep chamfer margin and 4~10 degrees of axial inclination using a diamond rotary bur. For the 3-unit bridge, a right mandibular first molar and premolar were prepared. The impression was obtained using a polysiloxane silicone impression material (Honigum-Light, DMG, Hamburg, Germany), which was cast with Type IV hard stone (Fujirock EP, GC Corp., Tokyo, Japan) to obtain the master cast.

The prepared abutment tooth working model was scanned using a 3D model scanner (Freedom HD, DOF, Seoul, Republic of Korea), which was then converted into an STL file format. The scanned 3D model was processed with a dental CAD program (Exocad, exocad GmbH, Darmstadt, Germany) to decide the margin and was designed using the conventional PFM restoration coping, with a cement gap of 30 μm.

The designed data were proposed to facilitate SLM Co-Cr alloy coping with a 3D printer (ProX^®^ DMP 100, 3D systems Inc., Rock Hill, SC, USA) using the SLM technique. The milling Co-Cr alloy coping was facilitated with the milling machine (Arden 5X-WM, TPS Korea LTD, Gwang-Ju, Republic of Korea). The Co-Cr alloy coping using the casting technique was achieved using the lost-wax technique. All of the procedures were undertaken by a single experienced laboratory technician in a uniform fashion. Ten Co-Cr alloy copings (single crown and a 3-unit bridge) were made for each of the 3 techniques, using the right mandibular first molar as an abutment (*n* = 10).

Porcelain was fused to the metal restoration specimens of all groups according to the manufacturer’s instructions. Noritake Super Porcelain EX-3 (Kuraray Noritake Dental Inc., Tokyo, Japan) was used for the porcelain build-up, and a porcelain furnace (Austromat 3001, Dekema, Freilassing, Germany) was used in accordance with the manufacturer’s instructions. (Figure 8) The porcelain build-up was held in 5 stages of ‘oxidizing–opaque–dentin–glazing’, with each stage including the first state, which was the ‘initial’ stage (Figure 8).

The marginal fit of the Co-Cr alloy coping determined with 3 different techniques was measured using metallurgic microscopy (EGVM-452M, Easytech, An-yang, Republic of Korea) at ×300 magnification following the marginal fit defined by Holmes [25]. The measurement was made at the center of the marginal buccal, lingual, mesial, and distal areas. The margin fit was measured at every 5 stages of porcelain firing, and each measurement was made at the same magnification and location and using the same method (Figure 9). Every group was measured with 10 specimens per measurement area, and the mean value of these was determined as the marginal fit of the respective area.

The internal fit was measured in all groups of specimens finished until the glazing stage of the porcelain firing procedures to compare the internal fit based on the different manufacturing techniques. The silicone replica technique, which is widely used in many studies and proven to be credible and accurate, was used to determine the internal space measurement [50,51]. The internal surface of each specimen was filled with polysiloxane silicone impression material (Honigum-Light, DMG, Hamburg, Germany) to determine the internal space. Each specimen filled with soft silicone material was mounted on the corresponding abutment models, which were then pressured in the direction of the tooth’s longitudinal axis. After the silicone was set, the specimens were removed, and a low-flowable polysiloxane silicone impression material (Honigum-Mono, DMG, Hamburg, Germany) was applied. After the final setting, the silicone was cut in a mesio–distal and linguo–buccal direction with a scalpel. The cross-section of the cut piece was observed at ×100 magnification using metallurgical microscopy (EGVM-452M, Easytech, An-yang, Republic of Korea), and the thickness of the silicone was measured to determine the internal gap (Figure 10). The internal gap of all specimens was measured in the mesio–distal and bucco–lingual part of the axial wall, cusp, and occlusal areas.

### 4.3. Statistical Analysis

In the case of the tensile strength test, the shear bond strength, and AFAP, a Kruskal–Wallis test was conducted to compare the mean values using the manufacturing techniques, followed by Mann–Whitney U test for post hoc analysis. In the case of the tests for Vickers microhardness, fitness, and surface roughness, A Shapiro–Wilk test was conducted to determine the normal distribution of the results. Following the test of normality, a one-way analysis of variance (ANOVA) was performed, followed by Tukey’s test for post hoc analysis. All of the statistical analyses were carried out using SPSS ver. 25.0 for Windows (SPSS Inc., Chicago, IL, USA).

## 5. Conclusions

In this study, a new method of manufacturing dental prosthetics using the SLM method was studied.

In the tensile strength test, the SLM technique exhibited the highest ultimate tensile strength and a 0.2% yield strength. This means that when the Co-Cr alloy specimen manufactured using the SLM technique is applied intra-orally, it has the lowest possibility of encountering permanent distortion due to masticatory force. Since the Co-Cr alloy using the SLM technique did not show a statistically significant difference in metal–porcelain bond strength compared to the conventional casting technique specimens, it is appropriate for clinical applications. The SLM technique exhibited better marginal fit compared to the conventional casting technique in all stages of porcelain firing procedures. This indicates that the Co-Cr alloy coping manufactured using the SLM technique can have more outstanding clinical performance. Co-Cr alloy coping manufactured using the SLM technique showed the highest internal gap.

Comprehensively compared to traditional casting methods, the dental prosthesis manufactured using the SLM method showed improved physical properties as well as improved marginal fit results, suggesting the possibility of clinical use.

## Figures and Tables

**Figure 1 ijms-24-07203-f001:**
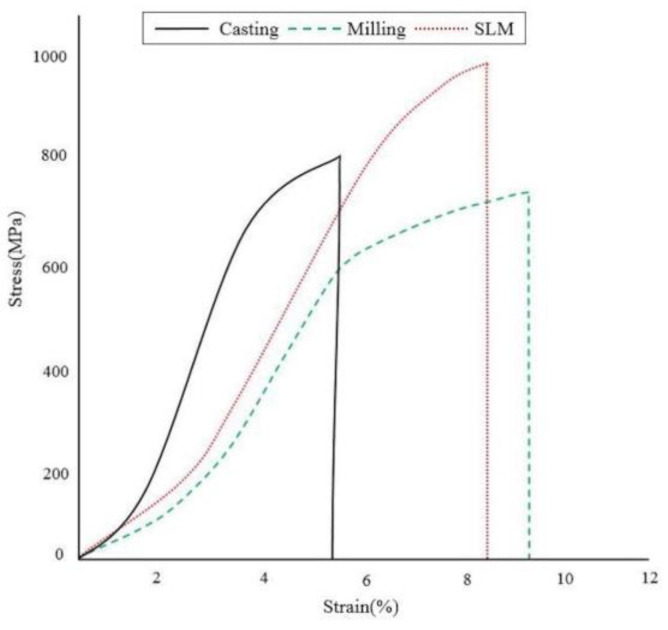
Stress–strain curves of tensile strength test following different manufacturing techniques.

**Figure 2 ijms-24-07203-f002:**
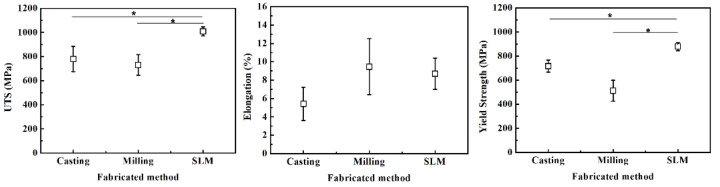
Ultimate tensile strength, elongation and yield strength following different manufacturing techniques. *, significant at *p* < 0.05.

**Figure 3 ijms-24-07203-f003:**
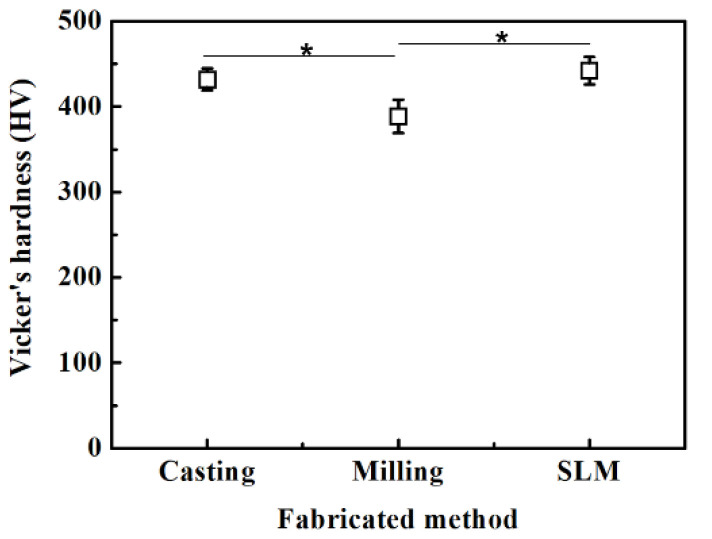
Mean ± SD Vicker’s microhardness values following different manufacturing techniques. *, significant at *p* < 0.05.

**Figure 4 ijms-24-07203-f004:**
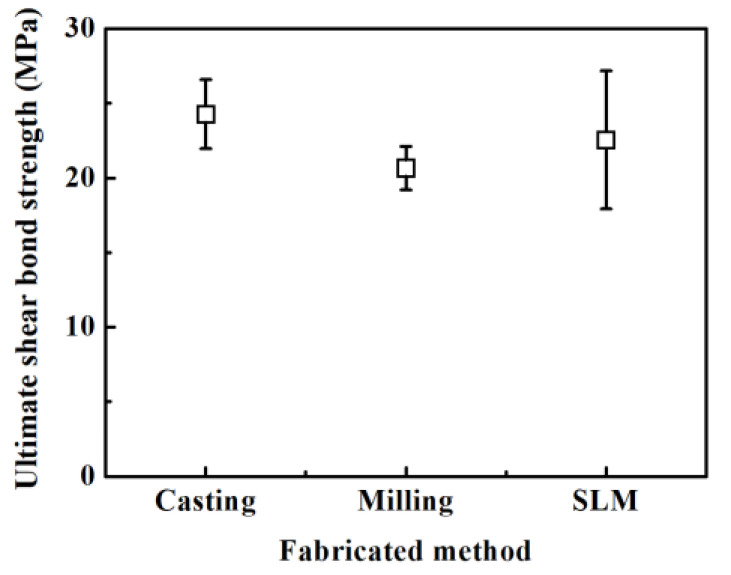
Shear bond strength values following different manufacturing techniques.

**Figure 5 ijms-24-07203-f005:**
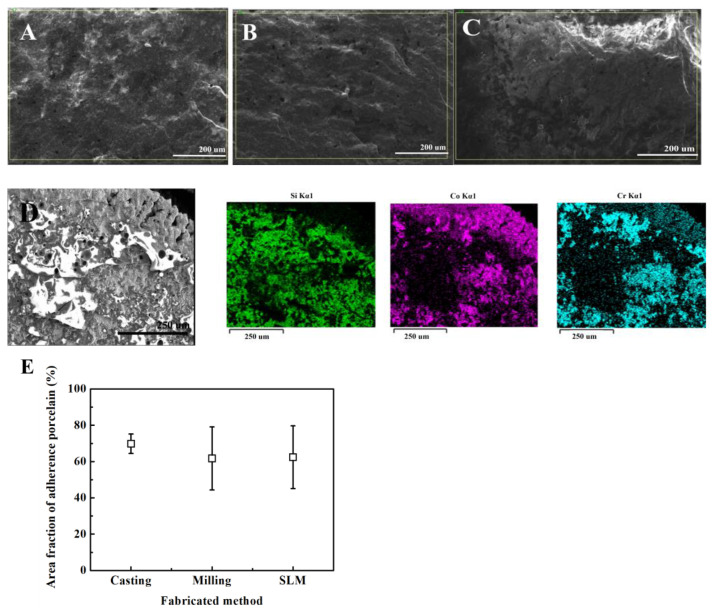
SEM and EDX results (×200 magnification) of casting, milling, SLM specimen metal–ceramic interface after fracture experiment. (**A**) casting, (**B**) milling, (**C**) SLM, (**D**) EDX mapping data on the metal–ceramic interface after fracture experiment. (**E**) area fraction of adherence porcelain values following different manufacturing techniques.

**Figure 6 ijms-24-07203-f006:**
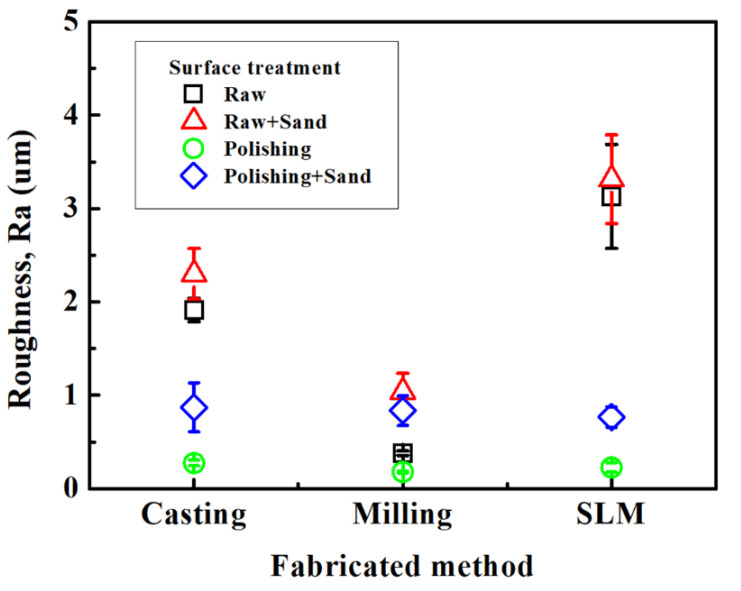
Mean ± SD surface roughness values (Ra) following different manufacturing techniques.

**Figure 7 ijms-24-07203-f007:**
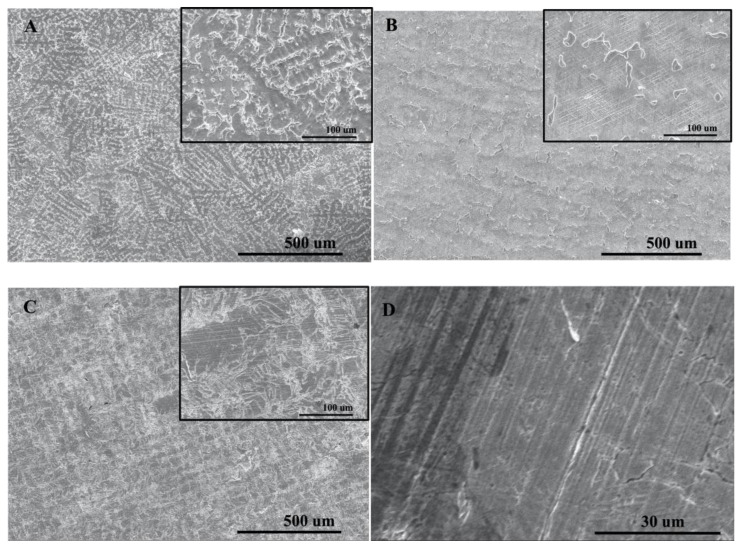
SEM images from polished section of Co-Cr alloys. (**A**) casting (original magnification × 100 and magnification of inserts × 500), (**B**) milling (original magnification × 100 and magnification of inserts × 500), (**C**) SLM (original magnification × 100 and magnification of inserts × 500). (**D**) SLM specimen (original magnification × 2000).

**Figure 8 ijms-24-07203-f008:**
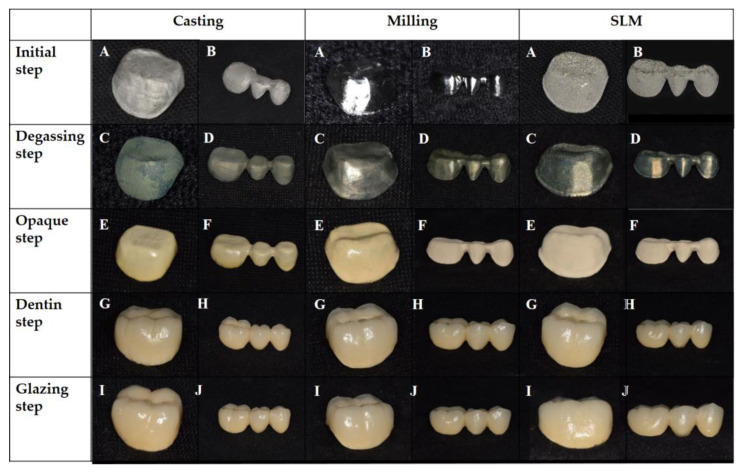
PFM dental restoration using casting, milling, and SLM methods. (**A**): single metal coping, (**B**): 3-unit bridge metal coping, (**C**): single coping after oxidizing process, (**D**): 3-unit bridge coping after the oxidizing process. (**E**): single coping after opaque porcelain firing procedure, (**F**): 3-unit bridge coping after opaque porcelain firing procedure, (**G**): single coping after dentin porcelain firing procedure, (**H**): 3-unit bridge coping after dentin porcelain firing procedure, (**I**): single coping after glazing procedure, (**J**): 3-unit bridge coping after glazing procedure.

**Figure 9 ijms-24-07203-f009:**
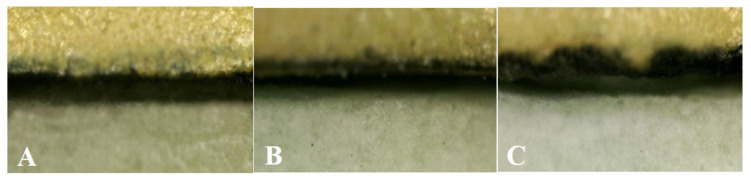
Marginal fit of PFM dental restoration after final glazing procedure at ×300 magnification (**A**) casting, (**B**) milling, (**C**) SLM.

**Figure 10 ijms-24-07203-f010:**
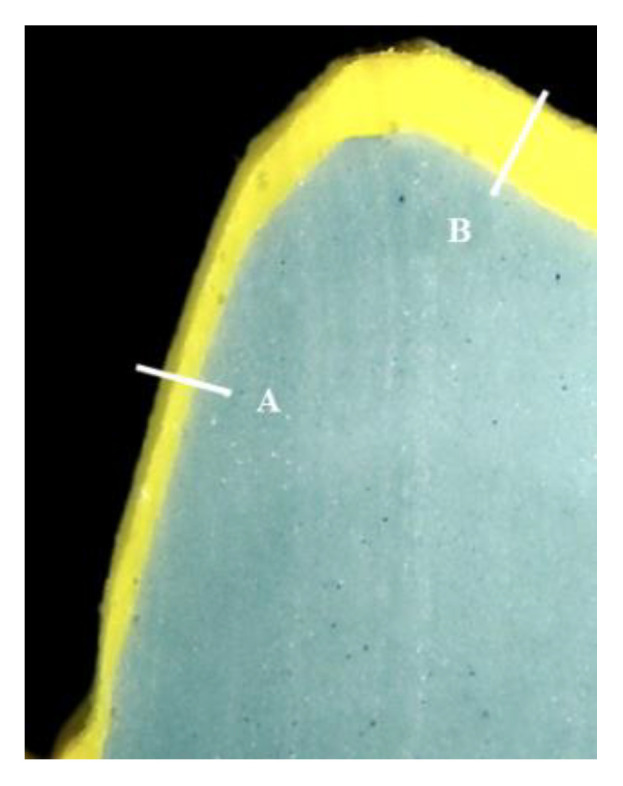
Internal fit of PFM dental single restoration at ×100 magnification. A: axial internal fit, B: occlusal internal fit.

**Table 1 ijms-24-07203-t001:** Marginal gap of casting, milling, SLM dental crown.

	Firing Cycle	Casting	Milling	SLM
Single crown	Initial	73.52 ± 13.86 ^e^	63.09 ± 17.46 ^a^	64.43 ± 13.47 ^a^
	Oxidizing	80.39 ± 13.86 ^i^	69.58 ± 13.53 ^b^	70.52 ± 13.46 ^c^
	Opaque	83.23 ± 14.24 ^j^	72.20 ± 19.86 ^e^	74.44 ± 18.03 ^f^
	Dentin	88.53 ± 11.84 ^l^	73.90 ± 17.51 ^e^	78.67 ± 18.87 ^g^
	Glazing	91.65 ± 8.71 ^l^	79.70 ± 16.40 ^h^	83.83 ± 17.38 ^j^
Three-unit bridge	Initial	81.05 ± 20.32 ^j^	67.55 ± 9.97 ^b^	71.62 ± 16.95 ^d^
	Oxidizing	88.04 ± 16.08 ^k^	75.68 ± 11.80 ^g^	77.63 ± 6.36 ^g^
	Opaque	91.65 ± 13.89 ^l^	79.80 ± 12.89 ^g^	83.76 ± 15.88 ^j^
	Dentin	96.09 ± 13.37 ^l^	82.05 ± 13.17 ^i^	86.45 ± 16.48 ^k^
	Glazing	100.07 ± 12.62 ^l^	86.59 ± 15.06 ^k^	92.64 ± 10.59 ^l^

The different letters represent a significant difference at a type-one error rate of 0.05.

**Table 2 ijms-24-07203-t002:** Internal gap of casting, milling, SLM single dental crown.

Firing Cycle	Casting	Milling	SLM
Axial	134.9 ± 28.29 ^a^	118.35 ± 24.26 ^b^	137.16 ± 31.94 ^a^
Occlusal	181.84 ± 28.68 ^d^	178.94 ± 26.22 ^d^	200.86 ± 22.16 ^c^
Total	160.98 ± 36.86 ^g^	152.01 ± 39.37 ^e^	172.55 ± 41.57 ^f^

The different letters represent a significant difference at a type-one error rate of 0.05.

**Table 3 ijms-24-07203-t003:** Internal gaps of a 3-unit bridge in casting, milling, SLM groups after the firing cycle process.

Internal Gap		Casting	Milling	SLM
First premolar	Axial	159.34 ± 10.02 ^c^	142.48 ± 12.32 ^f^	150.39 ± 9.39 ^b^
	Occlusal	179.3 ± 9.65 ^c,d,e^	186.42 ± 11.69 ^g^	205.07 ± 7.51 ^i^
First molar	Axial	159.07 ± 25.31 ^c,e^	141.26 ± 26.51 ^f^	154.85 ± 14.31 ^b^
	occlusal	201.54 ± 33.60 ^d^	201.29 ± 27.90 ^g^	246.74 ± 18.14 ^j^
total		174.82 ± 27.68 ^a^	167.86 ± 33.62 ^g^	189.26 ± 41.94 ^b^

The different letters represent a significant difference at a type-one error rate of 0.05.

**Table 4 ijms-24-07203-t004:** Chemical compositions of Co-Cr dental alloys.

Group	Co	Cr	Mo	Ma	Si	Fe	C	Product Name
Casting	64	28	6	<1	<1	<1	<1	Dental Alloy Products Super 6 Remelt Sources R516045 SinT-Tech ST2724G
Milling	Balance	27.9	5.79	0.35	0.83	0.30	0.24	
SLM	Balance	29	5.5	<1	<1	<1	<1	
ASTM F75-18 (Max)	Balance	30	7	1	1	0.75	0.35	
ASTM F75-18 (Min)	Balance	27	5	-	-	-	-	

## Data Availability

Not applicable.

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
