# Peer review of "Fundamental Properties and Clinical Application of 3D-Printed Bioglass Porcelain Fused to Metal Dental Restoration"

_ijms, 2023, doi:10.3390/ijms24087203_

Round 1

Reviewer 1 Report

The authors compared different techniques for the manufacturing of PMF restoration: LSM additive manufacturing technique, lost-wax casting, CAD/CAM milling. The results in terms of mechanical properties and fitting of the restoration were investigated. In my opinion, the work has a serious defect and cannot be accepted in this form. In fact, the cited literature is not uptaded, just two papers from 2017 are reported, while through fast research on Scopus, focusing on the year between 2017 and 2022, 17 papers related to this topic were found using the keywords:  internal; fit; dental; coping; cobalt. Several among them involve also the use of selective laser melting. The introduction must be updated with the last findings in this field, and the results obtained in this paper discussed again in this new light. More importantly, the novelty of the findings in this paper must be reassessed with the updated citations. Just then, and just in case of confirmed novelty and relevance, this paper may be considered for publication.

Overall, the paper can be improved also in the other section. See some comments below.

·        Fig 2: describe the meaning of *

·        Line 140: Correct Figure5 in Figure 5

·        Line 143: The authors say that the different phases, porcelain or metal, on the sample surfaces can be recognized by the contrast in the images. As I can see, the images are acquired using the secondary electrons, which do not provide compositional contrast, but topographical contrast instead. In my opinion, the light areas are due to the charging of the samples and it is not correct to locate porcelain in this way I suggest coating the samples with a thin metal layer (such Au or Pt) and perform back scattered images to highlight the different composition. Or EDS mapping can be suitable as well.

·        Line 156: which kind of surface roughness are you reporting? Is it Ra or Rq?

·        Figure 6: the representation of the statistical significance is not clear. Does the same letter over two columns means that there is statistical significance between them or there is significance between columns labelled with different letters? In either cases, I think that there are some inconsistencies in the representation of the statistical significance. The same comment applies also for Table 1,2 and 3.

·        Line 335 to 341: since there is no differences in the bond strength of samples prepared with different techniques, I think that it is not correct to attribute eventual differences to the thermal expansion coefficient, even in light of similar chemical composition and sample morphology. As consequence, the hypothesis that the thermal expansion coefficient is similar between the SLM and the casted samples cannot be sustained on this basis. I think that it would be better to measure it directly.

·        Line 431: the phrase is not clear, maybe chemical properties were intended instead of mechanical ones. Please rephrase.

·        Line 438: What is “CC”?

·        Line 451: how many measurements have been performed for the hardness measurement?

·        Line 458: how many samples have been tested, both for shear bond strength and for AFAP?

·        Line 467: the method applied to evaluate the AFAP by EDS, has been adapted from literature and validated in previous work? Because I think that the atm% of Si may not only depend on the area covered by the remaining porcelain but also on the thickness of the pieces that remain attached, which cannot be directly correlated to the covered area. Please clarify this point and report existing literature about this method.

·        Line 475: was the roughness of the samples evaluated according to ISO 4882? Which was the length of the profile and how many samples have been tested?

·        Fig 9: improve the figure by aligning the images and the stage of firing shall be described in the caption.

·        Fig. 10: improve the figure, in particular the picture in the last line. The meaning of the letters is missing in the caption.

·        Line 534: how many measurements of the internal gap, in both positions, were performed?

·        Line 552: Since a quite complex statistical analysis has been carried on, which is not so common in material science, I think that the rationale of these tests shall be described, along the parameters used and also the results of the various tests shall also be reported, maybe in supplementary information for instance.     

·        Conclusions: in the study, three manufacturing techniques were compared, but the milling process is not reported here. Please draw conclusion on all the content of the manuscript.

Generally, the image resolution shall be improved, and in SEM images the information and markers are not readable. They shall be enlarged, in particular the insets of figure 7. Tables format is not consistent through the text and Table 4 is an image instead of a table.  

Author Response

Response to Reviewer 1 Comments

Point 1: The authors compared different techniques for the manufacturing of PMF restoration: LSM additive manufacturing technique, lost-wax casting, CAD/CAM milling. The results in terms of mechanical properties and fitting of the restoration were investigated. In my opinion, the work has a serious defect and cannot be accepted in this form. In fact, the cited literature is not uptaded, just two papers from 2017 are reported, while through fast research on Scopus, focusing on the year between 2017 and 2022, 17 papers related to this topic were found using the keywords:  internal; fit; dental; coping; cobalt. Several among them involve also the use of selective laser melting. The introduction must be updated with the last findings in this field, and the results obtained in this paper discussed again in this new light. More importantly, the novelty of the findings in this paper must be reassessed with the updated citations. Just then, and just in case of confirmed novelty and relevance, this paper may be considered for publication.

Response 1: Thank you for your comments. References have been updated.

About 40 papers were searched on Scopus using the keyword: internal fit, SLM, porcelain. Most of them. Most of these papers were about removable prosthesis, not fixed prosthesis (PFM). Some papers measured the bonding strength or marginal fit of porcelain to Co-Cr alloy, but unlike the clinical situation, these papers replaced firing with thermocycling or only fired without raising porcelain. Marginal fitness has a great influence on the success of restorations in dental clinical situation. However, there are few papers on this and only papers that closely mimic single crowns. Generally, the longer the bridge, the worse the margin fit, but there is no paper on this. This paper is the only paper that has been identical to the clinical situation up to 3 unit bridge.

Point 2: Fig 2: describe the meaning of *

Response 2: Thank you for your comments. I added the meaning of * based on your comments.

Point 3: Line 140: Correct Figure5 in Figure 5

Response 3: Thank you for your comments. I revised it.

Point 4: Line 143: The authors say that the different phases, porcelain or metal, on the sample surfaces can be recognized by the contrast in the images. As I can see, the images are acquired using the secondary electrons, which do not provide compositional contrast, but topographical contrast instead. In my opinion, the light areas are due to the charging of the samples and it is not correct to locate porcelain in this way I suggest coating the samples with a thin metal layer (such Au or Pt) and perform back scattered images to highlight the different composition. Or EDS mapping can be suitable as well.

Response 4: Thank you for your kind comments.

The metal specimens after bond strength experiment was analyzed using SEM and EDX. In this experiment, there is no Si component in the metal, and the Si component is only in the porcelain. Thus we analyzed the SEM and EDX results according to the existence of Si (EDX) and AFAP results in the fractured metal surface.

Point 5: Line 156: which kind of surface roughness are you reporting? Is it Ra or Rq?

Response 5: Thank you for your kind comments. We used the Ra value. Thus we modified the manuscirpt and figure based on your comments.

Point 6: Figure 6: the representation of the statistical significance is not clear. Does the same letter over two columns means that there is statistical significance between them or there is significance between columns labelled with different letters? In either cases, I think that there are some inconsistencies in the representation of the statistical significance. The same comment applies also for Table 1,2 and 3.

Response 6: Thank you for your kind comments

The different letters represent a significant difference. Same alphabetic letters mean no statistical significance. Among the various ways to indicate statistical significance, letters were used because the graph is complicated to represent with *.

Although shown in the discussion, I want to briefly express the meaning of this graph (fig. 6) here. Immediately after fabrication of metal prostheses, cast and 3D printed specimens show higher roughness than milled specimens, which is statistically significant. When metal printing is performed, the surface is usually very rough, so the very rough part is cleaned using a dental bur. Sandblasting of these specimens shows a similar tendency. When all three specimens are polished, they show a similar level of roughness, and when sandblasting them, the roughness increases statistically significantly, but there is no significant difference between the manufacturing methods.

Point 7:  Line 335 to 341: since there is no differences in the bond strength of samples prepared with different techniques, I think that it is not correct to attribute eventual differences to the thermal expansion coefficient, even in light of similar chemical composition and sample morphology. As consequence, the hypothesis that the thermal expansion coefficient is similar between the SLM and the casted samples cannot be sustained on this basis. I think that it would be better to measure it directly.

I respect your opinion. It seems that there was an error in my thinking. I must apologize for disturbing you like this. I think this paragraph can be deleted.

Point 8: Line 431: the phrase is not clear, maybe chemical properties were intended instead of mechanical ones. Please rephrase.

Response 8: Thank you for your kind comments. Thank you for your kind comments.

I corrected that sentences. All specimen compositions for SLM, milling and casting techniques follow the ASTM F-75 standard so as to minimize the errors due to compositional differences in the materials

Point 9: Line 438: What is “CC”?

Response 9: Thank you for your kind comments.

I corrected that sentences. a Co-Cr Bar 

Point 10: Line 451: how many measurements have been performed for the hardness measurement?

We made 12 specimens for each groups to evalute the hardness. The hardness was measured once for each specimen, and since there were 12 specimens, the hardness was averaged over 12 values.

Point 11: Line 458: how many samples have been tested, both for shear bond strength and for AFAP?

We made 5 specimens for each groups to evaluate the bond strength expement and AFAP. It was measured once for each specimen, and since there were 5 specimens, the hardness was averaged over 5 values.

Point 12: Line 467: the method applied to evaluate the AFAP by EDS, has been adapted from literature and validated in previous work? Because I think that the atm% of Si may not only depend on the area covered by the remaining porcelain but also on the thickness of the pieces that remain attached, which cannot be directly correlated to the covered area. Please clarify this point and report existing literature about this method.

Point 13: Line 475: was the roughness of the samples evaluated according to ISO 4882? Which was the length of the profile and how many samples have been tested?

Response 13: Thank you for your kind comments.

Surface roughness was measured using 2D contact stylus porfilometer. Unfortunately We didn’t use the ISO 4882, but this profilometer (DIAVITE DH-8) machine complies with ISO 12085. The tracing length is 1.25mm.

Point 14:  Fig 9: improve the figure by aligning the images and the stage of firing shall be described in the caption.

Response 14: Thank you for your kind comments.

I modified the image and caption.  Marginal fit of PFM dental restoration after final glazing procedure (A) casting, (B) milling, (C) SLM

Point 15: Fig. 10: improve the figure, in particular the picture in the last line. The meaning of the letters is missing in the caption.

Response 15: Thank you for your kind commen. There are the meaning of the letters in the caption.

Point 16:  Line 534: how many measurements of the internal gap, in both positions, were performed?

Response 16: Thank you for your kind commen. We made 10 speciemens for each groups. It was measured once for each specimen, the gap was averaged over 10 values.

Point 17: Line 552: Since a quite complex statistical analysis has been carried on, which is not so common in material science, I think that the rationale of these tests shall be described, along the parameters used and also the results of the various tests shall also be reported, maybe in supplementary information for instance.    

Point 18:  Conclusions: in the study, three manufacturing techniques were compared, but the milling process is not reported here. Please draw conclusion on all the content of the manuscript.

I modified the conclusion based on your comments.

In this study, a new method of manufacturing dental prosthetics using the SLM method among 3D printing manufacturing methods was studied. Compared to the tradi-tional casting method, the dental prosthesis manufactured by the SLM method showed improved physical properties as well as improved marginal fit results, suggesting the possibility of clinical use.

In the tensile strength test, SLM technique showed highest ultimate tensile strength and 0.2% yield strength. This means that when the Co-Cr alloy specimen manufactured using SLM technique is applied intra-orally, it has the lowest possibility of encountering permanent distortion due to masticatory force. In the shear bond strength test and the area fraction of adherence porcelain (%), there were no statistically significant difference (p>0.05). This indicates that since Co-Cr alloy specimens manufactured using SLM tech-nique did not show statistically significant difference in metal-porcelain bond strength compared to the conventional casting technique specimens, it is appropriate for the clini-cal application.

In studying the change of marginal discrepancy values due to porcelain firing pro-cedures, all groups of Co-Cr alloy coping went through marginal discrepancy after going through a series of porcelain firing procedures (p<0.05). SLM technique showed better marginal fit compared to conventional casting technique in all stages of porcelain firing procedures. This indicates that Co-Cr alloy coping manufactured using SLM technique can have more outstanding clinical performance. Co-Cr alloy coping manufactured using SLM technique showed highest internal gap.

Comprehensively compared to the traditional casting method, the dental prosthesis manufactured by the SLM method showed improved physical properties as well as im-proved marginal fit results, suggesting the possibility of clinical use.

Point 19: Generally, the image resolution shall be improved, and in SEM images the information and markers are not readable. They shall be enlarged, in particular the insets of figure 7. Tables format is not consistent through the text and Table 4 is an image instead of a table.

Response to Reviewer 2 Comments

Point 1:  Regarding the References from the Introduction and the References in General, they are not new and some of them very old. I understand that there important and valuable information in the old articles too, but I  recommend to introduce some Updated References too.

I suggest some examples:

Jafari, N., Habashi, M.S., Hashemi, A. et al. Application of bioactive glasses in various dental fields. Biomater Res 26, 31 (2022). https://doi.org/10.1186/s40824-022-00274-6

Bioactive Glasses and Glass Ceramics: Fundamentals and Applications, 2022, Author Baino Francesco

Response 1: Thank you for your kind comments.

Based on your comments, I updated the reference. 

About 40 papers were searched on Scopus using the keyword: internal fit, SLM, porcelain. Most of them. Most of these papers were about removable prosthesis, not fixed prosthesis (PFM). Some papers measured the bonding strength or marginal fit of porcelain to Co-Cr alloy, but unlike the clinical situation, these papers replaced firing with thermocycling or only fired without raising porcelain. Marginal fitness has a great influence on the success of restorations in dental clinical situation. However, there are few papers on this and only papers that closely mimic single crowns. Generally, the longer the bridge, the worse the margin fit, but there is no paper on this. This paper is the only paper that has been identical to the clinical situation up to 3 unit bridge.

Thus I added some new reference in the introduction.

  1. hang M.; Gan N.; Qian H.; Jiao T. Retentive force and fitness accuracy of cobalt-chrome alloy clasps for removable partial denture fabricated with SLM technique J Prosthodont Res. 2022, 66,459-465.
  2. Daou E.E.; Özcan Evaluation of ceramic adherence to cobalt-chromium alloys fabricated by different manufacturing techniques. J Prosthet Dent. 2022, 128, 1364.e1-1364.e8.
  3. Alqahtani, A.S.; AlFadda, A.M.; Eldesouky, M.; Alnuwaiser, M.K.; Al-Saleh, S.; Alresayes, S.; Alshahrani, A.; Vohra, F.; Abduljabbar, T. Comparison of Marginal Integrity and Surface Roughness of Selective Laser Melting, CAD-CAM and Digital Light Processing Manufactured Co-Cr Alloy Copings. Sci. 2021, 11, 8328.

Thank you very much

Reviewer 2 Report

Dear authors, 

First of all congratulations for your work.

I find the subject very interesting, as the method of obtaining PFM restoration vi SLM technique is very new.

Regarding the References from the Introduction and the References in General, they are not new and some of them very old. I understand that there important and valuable information in the old articles too, but I  recommend to introduce some Updated References too.

I suggest some examples:

Jafari, N., Habashi, M.S., Hashemi, A. et al. Application of bioactive glasses in various dental fields. Biomater Res 26, 31 (2022). https://doi.org/10.1186/s40824-022-00274-6

Bioactive Glasses and Glass Ceramics: Fundamentals and Applications, 2022, Author Baino Francesco

Author Response

Response to Reviewer 2 Comments

Point 1:  Regarding the References from the Introduction and the References in General, they are not new and some of them very old. I understand that there important and valuable information in the old articles too, but I  recommend to introduce some Updated References too.

I suggest some examples:

Jafari, N., Habashi, M.S., Hashemi, A. et al. Application of bioactive glasses in various dental fields. Biomater Res 26, 31 (2022). https://doi.org/10.1186/s40824-022-00274-6

Bioactive Glasses and Glass Ceramics: Fundamentals and Applications, 2022, Author Baino Francesco

Response 1: Thank you for your kind comments.

Based on your comments, I updated the reference. 

About 40 papers were searched on Scopus using the keyword: internal fit, SLM, porcelain. Most of them. Most of these papers were about removable prosthesis, not fixed prosthesis (PFM). Some papers measured the bonding strength or marginal fit of porcelain to Co-Cr alloy, but unlike the clinical situation, these papers replaced firing with thermocycling or only fired without raising porcelain. Marginal fitness has a great influence on the success of restorations in dental clinical situation. However, there are few papers on this and only papers that closely mimic single crowns. Generally, the longer the bridge, the worse the margin fit, but there is no paper on this. This paper is the only paper that has been identical to the clinical situation up to 3 unit bridge.

Thus I added some new reference in the introduction.

  1. hang M.; Gan N.; Qian H.; Jiao T. Retentive force and fitness accuracy of cobalt-chrome alloy clasps for removable partial denture fabricated with SLM technique J Prosthodont Res. 2022, 66,459-465.
  2. Daou E.E.; Özcan Evaluation of ceramic adherence to cobalt-chromium alloys fabricated by different manufacturing techniques. J Prosthet Dent. 2022, 128, 1364.e1-1364.e8.
  3. Alqahtani, A.S.; AlFadda, A.M.; Eldesouky, M.; Alnuwaiser, M.K.; Al-Saleh, S.; Alresayes, S.; Alshahrani, A.; Vohra, F.; Abduljabbar, T. Comparison of Marginal Integrity and Surface Roughness of Selective Laser Melting, CAD-CAM and Digital Light Processing Manufactured Co-Cr Alloy Copings. Sci. 2021, 11, 8328.

Thank you very much

Round 2

Reviewer 1 Report

Point 1:

Since you have found about 40 papers on the topic, I suggest to add some more than just three, for example in line 79, to sustain your claim that Co-Cr was mainly used for removable prosthesis.

Point 4:

In my opinion, the authors do not address my point, either in their reply or in the manuscript. The EDX analysis is not reported in the manuscript, and from secondary electron images, it is not possible to say where a certain element is. To claim where is the porcelain and where is the metal, the authors shall include the chemical composition referred to a specific point of the surface.

 Point 16:

No changes in the text have been reported

Figures:
Generally, the quality of the figures is not improved. Figure 2 to 4 have low resolution; the scanning conditions and markers in Figure 5 are too small to be read, even by zooming in due to low resolution; Figure 6 name of y axes on top is missing; Figure 7 has very poor editing, if the images are too small for information to be seen, they shall be removed and only the markers left, without going outside the image and the same structure. The structure of composed images shall be similar through all the text. Figure 8 last line shall be improved, some letters are partially covered by the underneath images. Figure 9: the pictures are not aligned and the letters are too small.

The authors have improved the manuscript, still some important point must be addressed. Furthermore, the editing of the figure must be done more carefully and it is not of high enough quality for this journal currently. 

Author Response

Point 1:

Since you have found about 40 papers on the topic, I suggest to add some more than just three, for example in line 79, to sustain your claim that Co-Cr was mainly used for removable prosthesis.

Respond 1) Thank you for your kind comments.

References have been updated. I added the 6 reference that Co-Cr study for removable prosthesis.

Point 4:

In my opinion, the authors do not address my point, either in their reply or in the manuscript. The EDX analysis is not reported in the manuscript, and from secondary electron images, it is not possible to say where a certain element is. To claim where is the porcelain and where is the metal, the authors shall include the chemical composition referred to a specific point of the surface.

Respond 4) Thank you for your kind comments.

Since porcelain does not conduct electricity and metal does conduct electricity, it hastily concluded that the surface that looks bright due to the difference in conductivity must be porcelain, and only the element fraction of the EDX result was confirmed. But I agree with your opinion. Thus EDX mapping results are attached. Thank you for your opinion.

 Point 16:

No changes in the text have been reported

Respond 16) Thank you for your kind comments.

I apologize. It was written that 10 specimens were made on the contents, so it was not modified separately. This time, the number of specimens is indicated using parentheses. Thank you very much.

Figures:
Generally, the quality of the figures is not improved. Figure 2 to 4 have low resolution; the scanning conditions and markers in Figure 5 are too small to be read, even by zooming in due to low resolution; Figure 6 name of y axes on top is missing; Figure 7 has very poor editing, if the images are too small for information to be seen, they shall be removed and only the markers left, without going outside the image and the same structure. The structure of composed images shall be similar through all the text. Figure 8 last line shall be improved, some letters are partially covered by the underneath images. Figure 9: the pictures are not aligned and the letters are too small.

The authors have improved the manuscript, still some important point must be addressed. Furthermore, the editing of the figure must be done more carefully and it is not of high enough quality for this journal currently. 

 Respond figure) Thank you for your kind comments.

I have updated all figures. Figures 2 through 4 have all been redrawn.

The resolution of all pictures in Figure 5 has been increased. I also uploaded the original picture file separately in case it is difficult to see due to word space problems.

Figure 6 was also modified again.

Figure 7 was also modified. The resolution of Figure 7 has been increased, and the information of markers has been modified to appear in the figure.

Figure 8 was also modified based on your comments.

Figure 9 was also modified.

Round 3

Reviewer 1 Report

Dear authors, 

We have different opinions regarding how images shall be presented in a paper. So I just ask you to do a simple thing: download the PDF manuscript, set the magnification at 100% and check your images. If you deem them well readable and acceptable, so leave them as they are. In the other case, make the proper adjustment. 

Now the fig. 2 is missing the plot relative to the young's modulus that was in the previous version of the image. Please ammend it.

A brief comment on fig. 5: it is true that non-conductive materials are brighter than conductive materials, due to the charging effect. But if you have charging of the samples, then the EDX is not so reliable anymore and you shall coat your samples with Au or Pt for instance. 

kind regards

Author Response

Dear authors, 

We have different opinions regarding how images shall be presented in a paper. So I just ask you to do a simple thing: download the PDF manuscript, set the magnification at 100% and check your images. If you deem them well readable and acceptable, so leave them as they are. In the other case, make the proper adjustment. 

Respond 1) Thank you for your kind comments.

We apologize for bothering you. The resolution has been increased again. I also uploaded the original picture file separately again. Especially for SEM images, I added the information of markers in the images.

Now the fig. 2 is missing the plot relative to the young's modulus that was in the previous version of the image. Please ammend it.

Respond 2) Thank you for your kind comments.

In this paper, young's modulus was deleted during revision because it is not important in this paper and it is not a clinically important value.

A brief comment on fig. 5: it is true that non-conductive materials are brighter than conductive materials, due to the charging effect. But if you have charging of the samples, then the EDX is not so reliable anymore and you shall coat your samples with Au or Pt for instance. 

kind regards

Respond 3) Thank you for providing these insights.

When analyzing the fracture surface of metal and porcelain, the distribution or fraction of Si was considered important, so platinum coating was not performed. But I agree with you. We are continuing our research on metal printing. In the follow-up study, we will accept your opinion and perform surface analysis.

Again, thank you for giving us the opportunity to strengthen our manuscript with your valuable comments and queries.

Sincerely